# Identification of Bearing Dynamic Parameters and Unbalanced Forces in a Flexible Rotor System Supported by Oil-Film Bearings and Active Magnetic Devices

**Yinsi Chen** [1,2], **Ren Yang** [2], **Naohiro Sugita** [2], **Junhong Mao** [1] **and Tadahiko Shinshi** [2,*]

1    Key Laboratory of Education Ministry for Modern Design & Rotor-Bearing System, Theory of Lubrication and Bearing Research Institute, Xi'an Jiaotong University, Xi'an 710049, China; chen.y.bc@m.titech.ac.jp (Y.C.); jhmao@xjtu.edu.cn (J.M.)
2    Institute of Innovative Research, Tokyo Institute of Technology 4259 Nagatsuta-cho, Midori-ku, Yokohama 226-8503, Japan; yang.r.aa@m.titech.ac.jp (R.Y.); sugita.n.aa@m.titech.ac.jp (N.S.)
*    Correspondence: shinshi.t.ab@m.titech.ac.jp

**Abstract:** As the rotational speed of conventional rotor systems supported by oil-film bearings has increased, vibration problems such as oil whip and oil whirl have become apparent. Our group proposed the use of active magnetic bearings (AMBs)/bearingless motors (BELMs) to stabilize these systems. In such a system, measuring the variable stiffness and damping of the oil-film bearings, the current-force and displacement-force parameters of the AMBs/BELMs, and the residual unbalanced force is necessary to satisfy the stability of the rotor system. These parameters are the foundation for the rotor dynamics analysis and optimization of the control strategy. In this paper, we propose a method to simultaneously identify the parameters of the oil-film bearings and AMBs/BELMs along with the residual unbalanced forces during the unbalanced vibration of the rotor. The proposed method requires independent rotor responses and control currents to form a regression equation to estimate all the unknown parameters. Independent rotor responses are realized by changing the PID control parameters of the AMBs/BELMs. Numerical simulation results show that the proposed method is highly accurate and has good robustness to measurement noise. The experimental results show that the unknown parameters identified by the responses generated by different controller parameters are similar. To confirm that the identification results are correct, verification experiments were carried out. The vibration amplitude of the rotor was successfully suppressed by applying a force to the rotor in the opposite direction to the residual unbalanced force. The frequency response characteristics and unbalanced responses of the rotor estimated by the values of the parameters identified show good consistency with the measured results.

**Keywords:** parameter identification; rotating machinery; oil-film bearing; active magnetic bearing; bearingless motor; vibration control

## 1. Introduction

Oil-film bearings are widely used in rotating machinery systems, such as compressors and turbomachines. However, self-excited problems such as oil whirl and oil whip occur at high rotational speeds. With the increase in the rotational speed, the oil whirl occurs and turns to oil whip when the rotational speed is close to the first critical speed [1,2]. These fluid-induced vibrations produce large vibration amplitudes and decrease the stability of rotating machinery systems.

To suppress the self-excited vibrations, active magnetic bearings (AMBs) [3,4] and bearingless motors (BELMs) [5–7] have been used to reduce the vibration of the rotor-bearing systems. In such a rotor system supported by oil-film bearings combined with AMBs/BELMs, the stiffness and damping of the oil-film bearings and the current-force and displacement-force coefficients of the AMBs/BELMs are necessary for rotor dynamic

analysis, including the vibration response, the critical speeds, bending modes, and system stability. The current-force and displacement-force coefficients are also important for optimization of the control strategy since the stiffness and damping of the AMBs/BELMs are related to these coefficients and the feedback controller parameters.

Stiffness and damping parameters are the key dynamic parameters of oil-film bearings. These parameters have a substantial influence on the dynamic characteristics of the rotor. However, installation errors, misalignment, wear, and lubrication conditions have a significant effect on these parameters, which makes theoretical estimates of the bearing's dynamic parameters quite different from those involved in real rotor systems. Some studies have also used the stiffness and damping coefficients to express the dynamic characteristics of AMBs/BELMs similarly to traditional bearings. However, since the stiffness and damping parameters of AMBs/BELMs are related to the control parameters, the current–force and displacement–force coefficients are more suitable for describing the characteristics of AMBs/BELMs.

Furthermore, inaccuracies in manufacturing result in residual unbalance in the rotor. Installation errors, thermal deformation, and wear can change the balanced state of a balanced rotor. Excess forces generated by residual unbalanced lead to large vibrations, causing failure of the entire rotor system. Therefore, if the actual dynamic parameters of the oil-film bearings and those of the active magnetic device or the residual unbalanced involved in the rotor system are not known, the results of dynamic analysis are most likely to be incorrect.

Identification of the dynamic parameters of both conventional bearings and AMBs has been studied for many years. Incremental static load and dynamic excitation methods at both bearing components and the rotor-bearing system have been investigated [8–10]. With the incremental static load method, a load is applied to the rotor under stable working conditions. The measured parameters are stable and repeatable because there is no other external interference. However, the damping coefficients cannot be obtained because there is no transient force, and the identified parameters are highly sensitive to measurement errors [11,12]. Dynamic excitation methods are the most commonly used experimental methods. The stiffness and damping parameters of the bearing can be identified by applying dynamic forces to the bearing pedestal [13], but this method is not suitable for a full-scale rotating system. By contrast, applying the exciting force to the rotor shaft is more suitable for full-scale rotating systems identification [14–17]. These identification methods for all kinds of bearings have some similarities. The unknown parameters are calculated from the relationships between the input and output signals. The input signals are usually forces generated by unbalanced mass or exciter, and the output signals can be displacements, velocities, and accelerations.

Hydraulic excitation has been used to excite rotors [18,19]. However, to obtain the actual dynamic parameters, it is recommended that no additional parts are added or any mechanical connection made between the rotor and the measuring device because the additional components change the natural characteristics of the rotor. To avoid contact, electromagnetic excitation has become a convenient method for applying dynamic force [20,21]. Both electromagnetic exciter and AMB have been used for the identification of bearing parameters and test the stability of the rotor systems [22,23]. The difference between the electromagnetic exciter and AMB is that the structure of the electromagnetic exciter is simpler.

As well as these two excitation methods, the unbalanced mass method, which does not need extra devices and has high accuracy, has been widely used in different rotor systems. Although unbalanced excitation is simple, the disadvantage is that at least two test runs are necessary to estimate all eight dynamic coefficients [24].

Furthermore, synchronous unbalanced responses caused by residual imbalances, which can be easily obtained from the rotor system, should be investigated for identification. However, since the excitation frequency is synchronized with the rotational speed and the position of the excitation force cannot be changed, it is difficult to produce linearly

independent rotor responses at the same speed for identification. However, with the help of an active magnetic device, the rotor dynamics can be changed actively, and different responses of the rotor can be achieved by changing the closed-loop control parameters at the same speed. This provides the possibility of identification and was adopted in our study.

In this paper, to simultaneously identify the stiffness and damping of the oil-film bearings, the current–force and displacement–force coefficients of the AMBs/BELMs, and the residual unbalanced force, we proposed an identification method based on the measured rotor unbalanced responses and control currents under different PID controller coefficients. This method has not been used or studied in earlier works.

The proposed method required independent rotor responses and control currents to form a regression equation to estimate all the unknown parameters. The unknown residual unbalanced force at a predefined plane is regarded as the excitation force, and different displacement responses of the oil-film bearings and AMBs/BELMs and the control currents of the AMBs/BELMs can be obtained by changing the PID control parameters of the AMBs/BELMs. The finite element (FE) method is used to build the numerical model of the flexible rotor. The amplitude and phase information of the control currents and the rotor responses are obtained by Fast Fourier Transforms (FFT). The rest of this paper is organized as follows. Section 2 describes the mathematical modeling of the rotor system and the assumptions made regarding the model. In Section 3, we describe the identification method. In Section 4, the numerical simulation results are given, including the robustness to measurement noise. Section 5 described the experimental conditions and verification of the identification results. Section 6 gives the conclusions.

## 2. System Modeling

A flexible rotor supported on two circular oil-film bearings and a BELM is considered for the study, as shown in Figure 1. The BELM is a combination of a radial AMB and a motor, which can generate the radial magnetic force and the rotational torque for rotor suspension and rotation.

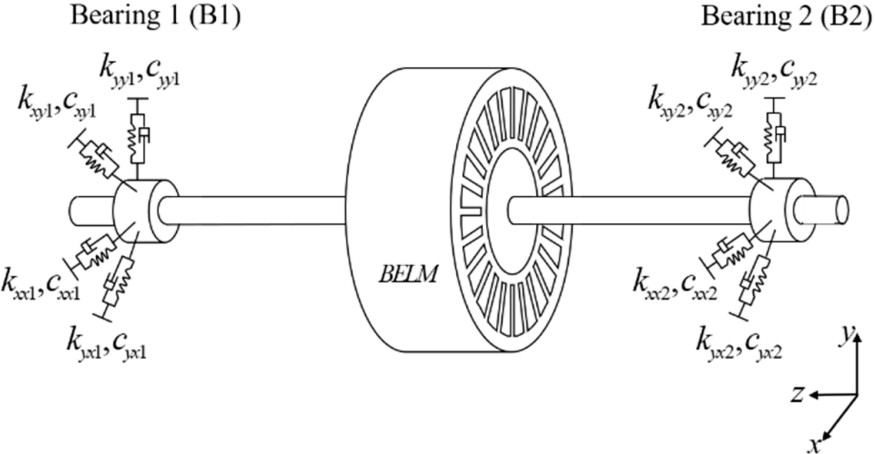

**Figure 1.** BELM rotor-bearing system.

The rotor is modeled by the Timoshenko beam theory. The gyroscopic effects, shear deformation, and rotatory inertia are considered in the finite element model. The assembled parts on the rotor are modeled as rigid discs. The damping of the shaft is ignored because it is negligibly small compared to the damping effect of the oil-film bearings and the BELM.

For the shaft element, the equation of motion of the rotor system is

$$\mathbf{M}_e \ddot{\mathbf{q}}_e + \Omega \mathbf{G}_e \dot{\mathbf{q}}_e + \mathbf{K}_e \mathbf{q}_e = \mathbf{F}_e \tag{1}$$

with

$$\mathbf{q}_e = \left\{ \begin{array}{cccc} x_e & y_e & \theta_{xe} & \theta_{ye} \end{array} \right\}^T$$

where $\mathbf{q}_e$ is the elemental nodal displacement vector; $x_e$ and $y_e$ are translations in the $x$ and $y$ directions; $\theta_{xe}$ and $\theta_{ye}$ are the angular displacements in the $x$ and $y$ directions; $\mathbf{F}_e$ is the elemental nodal force vector; $\mathbf{M}_e$, $\mathbf{K}_e$, and $\mathbf{G}_e$ are elemental mass, stiffness, and gyroscopic matrices; $\Omega$ is the rotor speed.

The BELM is considered to have linearized displacement–force and current–force coefficients in the $x$ and $y$ directions. The controlling force generated by the suspension winding of the BELM at a particular node can be expressed as

$$\mathbf{F}_a = \mathbf{K}_s \mathbf{q}_a + \mathbf{K}_i \mathbf{I}_a \tag{2}$$

where

$$\mathbf{K}_s = \left[ \begin{array}{cc} k_{sx} & 0 \\ 0 & k_{sy} \end{array} \right], \quad \mathbf{K}_i = \left[ \begin{array}{cc} k_{ix} & 0 \\ 0 & k_{iy} \end{array} \right]$$

where $\mathbf{K}_s$ is the displacement–force parameter matrix; $\mathbf{K}_i$ is the current–force parameter matrix; $\mathbf{q}_a$ is the BELM displacement vector and $\mathbf{I}_a$ is the control current; $k_{sx}$ and $k_{sy}$ represent the displacement–force parameters in the $x$ and $y$ directions; $k_{ix}$ and $k_{iy}$ represent the current–force parameters in the $x$ and $y$ directions.

For the oil-film bearing, the linearized oil-film bearing force can be expressed as

$$\mathbf{F}_b = -\mathbf{K}_b \mathbf{q}_b - \mathbf{C}_b \dot{\mathbf{q}}_b \tag{3}$$

where

$$\mathbf{K}_b = \left[ \begin{array}{cc} k_{xx} & k_{xy} \\ k_{yx} & k_{yy} \end{array} \right], \mathbf{C}_b = \left[ \begin{array}{cc} c_{xx} & c_{xy} \\ c_{yx} & c_{yy} \end{array} \right]$$

where $\mathbf{K}_b$ and $\mathbf{C}_b$ are the stiffness and damping matrices of the bearing; $\mathbf{q}_b$ is the bearing displacement vector; $k_{xx}$ and $k_{yy}$ are the direct stiffness parameters of the bearing in the $x$ and $y$ directions; $k_{xy}$ and $k_{yx}$ are the cross-coupling stiffness parameters of the bearing; $c_{xx}$ and $c_{yy}$ are the direct damping parameters; $c_{xy}$ and $c_{yx}$ are the cross-coupling damping parameters.

The equations of motion of the whole rotor system can be expressed by combining the equations for all the elements

$$\mathbf{M}_R \ddot{\mathbf{q}}_R + \Omega \mathbf{G}_R \dot{\mathbf{q}}_R + \mathbf{K}_R \mathbf{q}_R = \mathbf{F}_u + \mathbf{F}_a - \mathbf{F}_b \tag{4}$$

where $\mathbf{M}_R$, $\mathbf{K}_R$, and $\mathbf{G}_R$ are the global mass, stiffness, and gyroscopic matrices; $\mathbf{q}_R$ is the generalized displacement vector; $\mathbf{F}_u$ is the unbalanced force vector; $\mathbf{F}_a$ and $\mathbf{F}_b$ are the BELM controlling force and bearing force vector, respectively.

## 3. Identification Algorithm

Since the number of measuring points in the practical rotor system is less than the number of displacement vectors in the FE model, the dynamic reduction method is used to reduce the DOFs in the FE equation. The transformation matrix is defined as

$$\mathbf{q}_R = \left\{ \begin{array}{c} \mathbf{q}_m \\ \mathbf{q}_s \end{array} \right\} = \mathbf{T}^r \mathbf{q}_m \tag{5}$$

where

$$\mathbf{T}^r = \left[ \begin{array}{c} \mathbf{I} \\ -\mathbf{K}_{ss}^{-1} \mathbf{K}_{sm} \end{array} \right] \tag{6}$$

where $\mathbf{T}^r$ is the dynamic reduction transformation matrix. The subscripts *m* and *s* are the measured and unmeasured DOFs, respectively. After the dynamic reduction, Equation (4) can be rewritten as

$$\mathbf{M}^r \ddot{\mathbf{q}}_m + \Omega \mathbf{G}^r \dot{\mathbf{q}}_m + \mathbf{K}^r \mathbf{q}_m = \mathbf{F}^r \tag{7}$$

with

$$\begin{aligned} \mathbf{M}^r &= (\mathbf{T}^r)^T \mathbf{M}_R \mathbf{T}^r, \quad \mathbf{K}^r = (\mathbf{T}^r)^T \mathbf{K}_R \mathbf{T}^r \\ \mathbf{G}^r &= (\mathbf{T}^r)^T \mathbf{G}_R \mathbf{T}^r, \quad \mathbf{F}^r = (\mathbf{T}^r)^T (\mathbf{F}_u + \mathbf{F}_A - \mathbf{F}_B) \end{aligned} \tag{8}$$

where $\mathbf{q}_m$ is the measured rotor displacement vectors; $\mathbf{F}^r$ is the force vectors; and $\mathbf{M}^r$, $\mathbf{K}^{r\prime}$ and $\mathbf{G}^r$ are the reduced mass, stiffness, and gyroscopic matrices, respectively.

In general, $\mathbf{F}^r$ can be expressed as a combination of a forward and backward rotating force

$$\mathbf{F}^r = \mathbf{F}_A e^{j\Omega t} + \mathbf{F}_B e^{-j\Omega t} \tag{9}$$

where $\mathbf{F}_A$ and $\mathbf{F}_B$ are the force magnitudes in the complex form of the forward and backward excitations. Note that an excitation force with angular frequency $\omega$ caused by the unbalanced force is synchronized with the rotating speed, $\omega = \Omega$. Therefore, the displacement response and the control current have the same form as the excitation

$$\begin{aligned} \mathbf{q}_m &= \mathbf{q}_A e^{j\Omega t} + \mathbf{q}_B e^{-j\Omega t} \\ \mathbf{I}_a &= \mathbf{I}_A e^{j\Omega t} + \mathbf{I}_B e^{-j\Omega t} \end{aligned} \tag{10}$$

According to Equations (7), (9) and (10) can be rewritten as

$$\left. \begin{aligned} (-\omega^2 \mathbf{M}^r + j\omega\Omega \mathbf{G}^r + \mathbf{K}^r)\mathbf{q}_A &= \mathbf{F}_A \\ (-\omega^2 \mathbf{M}^r - j\omega\Omega \mathbf{G}^r + \mathbf{K}^r)\mathbf{q}_B &= \mathbf{F}_B \end{aligned} \right\} \tag{11}$$

In Equation (11), we can arrange all the unknown parameters (i.e., dynamic parameters of the BELM, the oil-film bearing dynamic parameters, and the residual unbalanced force) in a vector. The measured control currents, the rotor responses, and the parameters of the rotor model can be arranged as a regression matrix. The reordered equation can be expressed as

$$\mathbf{H} = [\mathbf{A}]\{\mathbf{X}\} \tag{12}$$

with

$$\begin{aligned} \mathbf{H} &= \left[ \begin{array}{c} (-\omega^2 \mathbf{M}^r + j\omega\Omega \mathbf{G}^r + \mathbf{K}^r)\mathbf{q}_A \\ (-\omega^2 \mathbf{M}^r - j\omega\Omega \mathbf{G}^r + \mathbf{K}^r)\mathbf{q}_B \end{array} \right] \\ \mathbf{X} &= \left\{ \begin{array}{ccccc} \mathbf{k}_b & \mathbf{c}_b & \mathbf{k}_s & \mathbf{k}_i & \mathbf{f}_u \end{array} \right\}^T \end{aligned}$$

where $\mathbf{A}$ is the regression matrix; $\mathbf{X}$ is the vector that contains all the unknown parameters; and $\mathbf{k}_b$, $\mathbf{c}_b$, $\mathbf{k}_s$, $\mathbf{k}_i$, and $\mathbf{f}_u$ are the bearing stiffness, the bearing damping, the BELM displacement-force parameter, the BELM current-force parameter, and the residual unbalanced force vector, respectively.

However, the number of unknown parameters is more than the measured DOFs. To obtain a sufficient number of equations to derive the unknown parameters, the BELM control parameters are changed to produce an adequate number of data sets for the rotor responses.

Take the rotor system in Figure 1 as an example. The residual unbalanced force is generated on the BELM plane when the rotor is rotated. A proportional-integral-derivative (PID) controller is used for rotor levitation, and the controller transfer function *G*(s) is

$$G(s) = P + \frac{I}{s} + D\frac{N_f s}{N_f + s} \tag{13}$$

where *P*, *I*, and *D* are the proportional, integral, and derivative coefficients; $N_f$ is the filter coefficient.

According to Equation (12), the unknown vectors of the rotor system $\mathbf{X}_{24\times1}$ has the following elements

$$
\mathbf{k}_b = \left\{ \begin{array}{c} k_{xxi} \\ k_{xyi} \\ k_{yxi} \\ k_{yyi} \\ \vdots \end{array} \right\}_{i=1,2}; \mathbf{c}_b = \left\{ \begin{array}{c} c_{xxi} \\ c_{xyi} \\ c_{yxi} \\ c_{yyi} \\ \vdots \end{array} \right\}_{i=1,2}; \mathbf{k}_s = \left\{ \begin{array}{c} k_{sx} \\ k_{sy} \end{array} \right\}; \mathbf{k}_i = \left\{ \begin{array}{c} k_{sx} \\ k_{sy} \end{array} \right\}; \mathbf{f}_u = \left\{ \begin{array}{c} f^r{}_x \\ f^i{}_x \\ f^r{}_y \\ f^i{}_y \end{array} \right\}
$$

(14)

where the subscript $i$ is the number of the bearing. The superscripts $r$ and $i$ are the real and imaginary parts of the residual unbalanced force.

Two sets of PID controller parameters are used to produce 24 equations, the overall estimation equation can be expressed as

$$
\left[ \begin{array}{c} \mathbf{H}_1 \\ \mathbf{H}_2 \end{array} \right]_{24\times1} = \left[ \begin{array}{c} \mathbf{A}_1 \\ \mathbf{A}_2 \end{array} \right]_{24\times24} \mathbf{X}_{24\times1}
$$

(15)

where

$$
\mathbf{H}_i = \left[ \begin{array}{c} (-\omega^2\mathbf{M}^d + j\omega\Omega\mathbf{G}^d + \mathbf{K}^d)\mathbf{q}_{Ai} \\ (-\omega^2\mathbf{M}^d - j\omega\Omega\mathbf{G}^d + \mathbf{K}^d)\mathbf{q}_{Bi} \end{array} \right]_{i=1,2}
$$

where the elements of the regression matrix $\mathbf{A}_1$ are

$A_{1,1} = A_{2,3} = -B_{1xA}$, $A_{1,2} = A_{2,4} = -B_{1yA}$, $A_{1,5} = A_{2,7} = -j\Omega B_{1xA}$, $A_{1,6} = A_{2,8} = -\Omega B_{1yA}j$

$A_{3,17} = X_A$, $A_{3,18} = I_{xA}$, $A_{3,21} = 0.5$, $A_{3,22} = -0.5j$

$A_{4,19} = Y_A$, $A_{4,20} = I_{yA}$, $A_{4,23} = 0.5$, $A_{4,24} = -0.5j$

$A_{5,9} = A_{6,11} = -B_{2xA}$, $A_{5,10} = A_{6,12} = -B_{2yA}$, $A_{5,13} = A_{6,15} = -\Omega B_{2xA}j$, $A_{5,14} = A_{6,16} = -\Omega B_{2yA}j$

$A_{7,1} = A_{8,3} = -B_{1xB}$, $A_{7,2} = A_{8,4} = -B_{1yB}$, $A_{7,5} = A_{8,7} = \Omega B_{1xB}j$, $A_{7,6} = A_{8,8} = \Omega B_{1yB}j$

$A_{9,17} = X_B$, $A_{9,18} = I_{xB}$, $A_{9,21} = 0.5$, $A_{9,22} = 0.5j$

$A_{10,19} = Y_B$, $A_{10,20} = I_{yB}$, $A_{10,23} = 0.5$, $A_{10,24} = 0.5j$

$A_{11,9} = A_{12,11} = -B_{2xB}$, $A_{11,10} = A_{12,12} = -B_{2yB}$, $A_{11,13} = A_{12,15} = \Omega B_{2xB}j$, $A_{11,14} = A_{12,16} = \Omega B_{2yB}j$

where $X_A$ and $Y_A$ are the forward parts of the displacement response amplitude of the BELM in the $x$ and $y$ directions; $B_{1xA}$, $B_{1yA}$, $B_{2xA}$, and $B_{2yA}$ are the forward parts of the displacement response amplitude of bearing1 and bearing 2 in the $x$ and $y$ directions; $I_{xA}$ and $I_{yA}$ are the forward parts of the control current in the $x$ and $y$ directions; $X_B$, $Y_B$, $B_{1xB}$, $B_{1yB}$, $B_{2xB}$, $B_{2yB}$, $I_{xB}$, and $I_{yB}$ are the corresponding amplitudes of the backward parts. In the matrix, other elements are zero. Here, we only give the formulas for $\mathbf{A}_1$; however, $\mathbf{A}_2$ has the same format as $\mathbf{A}_1$, but the response data are different. The unknown parameters of the system can be determined by solving Equation (15).

The PID control parameters should be selected to produce linearly independent equations. In a closed-loop suspension control system, the proportional coefficient of the PID controller affects the stiffness of the system. Therefore, different rotor responses can be obtained by changing just the *P* coefficient. The two linearly independent responses can be obtained by changing the ratio of the *P* coefficient in the $x$ and $y$ directions. Moreover, although the small cross-coupling of the BELM can be negligible, inappropriate selection of the coefficient may influence the characteristics of the oil-film bearing in the case of eccentric rotation. To eliminate the effect of cross-coupling on the dynamic parameters of the oil-film bearings, the change in *P* coefficient in both the $x$ and $y$ directions should be the same. Thus, the selection of *P* coefficients should obey the following equations

$$
\left\{ \begin{array}{l} \dfrac{P_{x2}}{P_{y2}} = \dfrac{P_{x1}\pm\delta}{P_{y1}\pm\delta} \\ \dfrac{P_{x1}}{P_{y1}} \neq \dfrac{P_{x2}}{P_{y2}} \end{array} \right.
$$

(16)

where $P_{x1}$ and $P_{y1}$ are the first set of the proportional coefficients of the PID controller in the $x$ and $y$ directions; $P_{x2}$ and $P_{y2}$ are the second set; $\delta$ is a variable.

## 4. Numerical Simulations

In this section, the accuracy and the anti-noise capability of the method were evaluated in a rotor system model, as shown in Figure 1.

The numerical rotor model is compiled in SIMULINK to obtain the BELM control current and rotor displacement responses in the $x$ and $y$ directions. Table 1 shows the physical properties of the BELM rotor-bearing system. The procedure for numerical evaluation is shown in Figure 2. The block diagram of the BELM rotor-bearing system is shown in Figure 3. The residual unbalanced force is regarded as the excitation force added onto the BELM node. The simulated rotational speed is 2100 rpm. The equations are solved by the Runge–Kutta method, and the fundamental sampling time of the solver is 1/20000s. The steady-state amplitude and phase information of the rotor responses are obtained by fast Fourier transforms (FFT) because the signal-to-noise ratio is found to be better in the frequency domain. The PID control parameters used for simulation are shown in Table 2.

**Table 1.** Specifications of the rotor system.

| Components | Parameters | Values |
|---|---|---|
| Shaft | Length | 780 mm |
| | Diameter | $\Phi 15$ mm |
| | Young's modulus | $2.05 \times 1011$ N/m$^2$ |
| | Mass density | 7850 kg/m$^3$ |
| | Poisson's ratio | 0.29 |
| Oil-film bearing | Length | 25 mm |
| | Diameter | 25 mm |
| Motor | Rotor length | 50 mm |
| | Rotor diameter | 70 mm |

**Table 2.** PID control parameters.

| Directions | Parameters | First Set | Second Set |
|---|---|---|---|
| $x$-direction | Proportional ($P$) | 52,000 | 54,000 |
| | Integral ($I$) | 17,500 | 17,500 |
| | Derivative ($D$) | 60 | 60 |
| | Filter coefficient ($N_f$) | 12,560 | 12,560 |
| $y$-direction | Proportional ($P$) | 54,000 | 52,000 |
| | Integral ($I$) | 17,500 | 17,500 |
| | Derivative ($D$) | 60 | 60 |
| | Filter coefficient ($N_f$) | 12,560 | 12,560 |

According to the flow chart, the unknown parameters are estimated based on the simulated rotor unbalanced responses and control currents under different PID controller coefficients. Table 3 gives the assumed and estimated stiffness and damping parameters of the oil-film bearings. Similarly, comparisons of the assumed and estimated displacement–force coefficient, current–force coefficient, and residual unbalanced force are listed in Table 4.

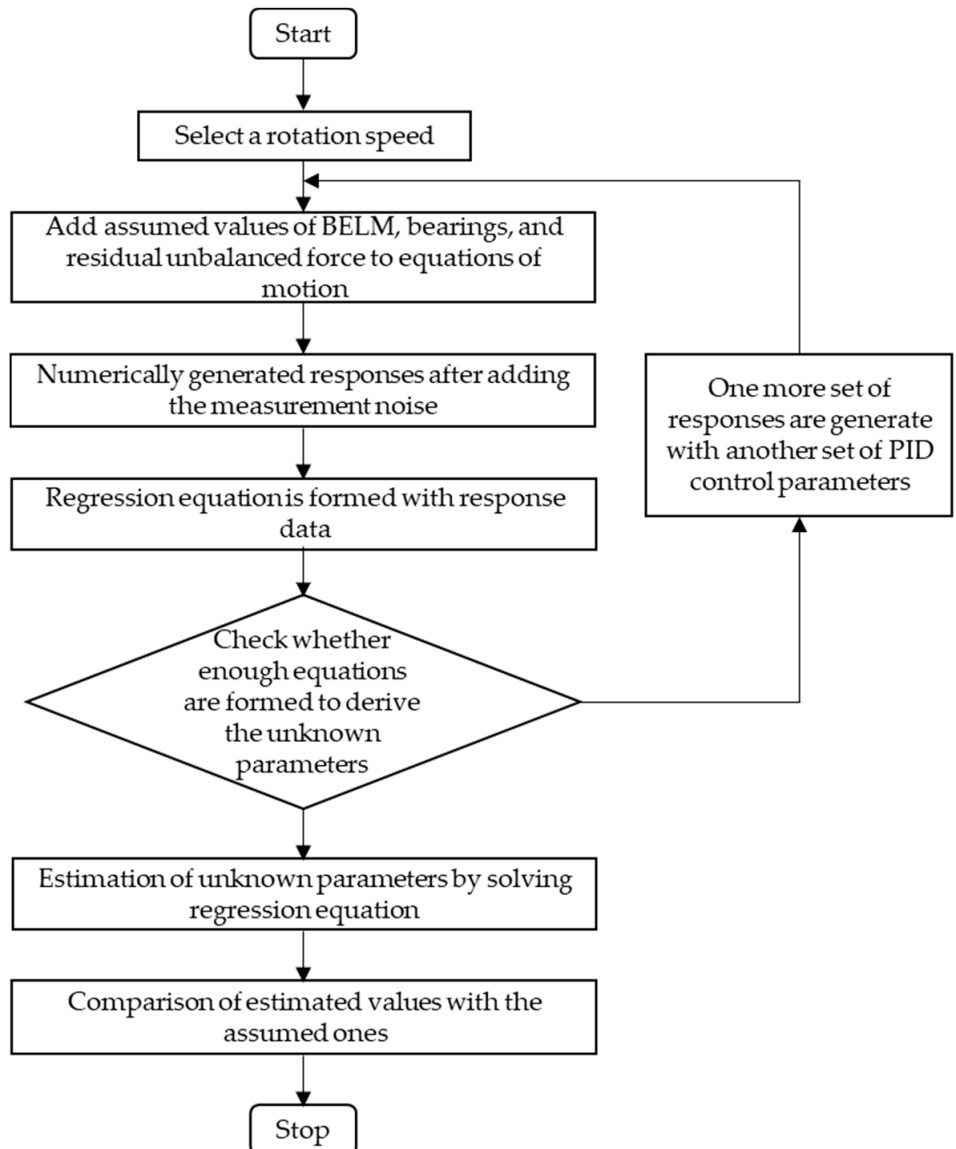

**Figure 2.** Flow chart of the unknown parameters identification.

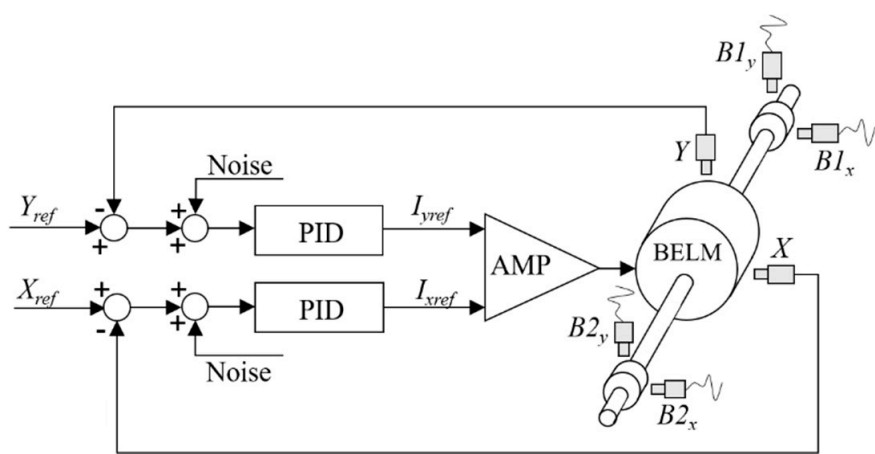

**Figure 3.** Block diagram for the suspension control and response measurement of the rotor system.

**Table 3.** Assumed and estimated stiffness and damping parameters of the two bearings.

| Parameters | | Bearing 1 | | Bearing 2 | |
|---|---|---|---|---|---|
| | | Assumed | Estimated | Assumed | Estimated |
| Stiffness (N/m) | $k_{xx}$ | $1 \times 10^5$ | $1.000 \times 10^5$ | $1.4 \times 10^5$ | $1.399 \times 10^5$ |
| | $k_{xy}$ | $1.2 \times 10^5$ | $1.200 \times 10^5$ | $1.6 \times 10^5$ | $1.599 \times 10^5$ |
| | $k_{yx}$ | $1.4 \times 10^5$ | $1.399 \times 10^5$ | $1.8 \times 10^5$ | $1.800 \times 10^5$ |
| | $k_{yy}$ | $1.8 \times 10^5$ | $1.799 \times 10^5$ | $2.0 \times 10^5$ | $2.000 \times 10^5$ |
| Damping (N.s/m) | $c_{xx}$ | 1000 | 1000.000 | 1200 | 1199.999 |
| | $c_{xy}$ | 800 | 800.000 | 500 | 499.999 |
| | $c_{yx}$ | 600 | 599.999 | 600 | 600.000 |
| | $c_{yy}$ | 2000 | 1999.999 | 1500 | 1500.000 |

**Table 4.** Assumed and estimated parameters of the BELM and the residual unbalanced force.

| Directions | Parameters | Assumed | Estimated |
|---|---|---|---|
| $x$-direction | $k_{sx}$ (N/m) | $8.62 \times 10^5$ | $8.619 \times 10^5$ |
| | $k_{ix}$ (N/A) | 28 | 29.999 |
| | $f^r{}_x$ (N) | 34.64 | 34.641 |
| | $f^i{}_x$ (N) | $-20$ | $-20.000$ |
| $y$-direction | $k_{sy}$ (N/m) | $8.72 \times 10^5$ | $8.719 \times 10^5$ |
| | $k_{iy}$ (N/A) | 28.8 | 28.799 |
| | $f^r{}_y$ (N) | 25 | 24.999 |
| | $f^i{}_y$ (N) | 43.30 | 43.301 |

In a real system, measurement noise is unavoidable. To evaluate the anti-noise ability, 1%, 3%, 5%, and 10% of noise are added to the BELM and the bearings' responses. The simulated response with noise is expressed as

$$S_i = B_i + NB_A R_i, i = 1, 2, \ldots, n$$
$$N = 1\%, 3\%, 5\%, 10\%$$

where $B$ is the response signal; $B_A$ is the amplitude of the vibration; $R$ is a Gaussian random sequence in the range of $-1$ to $1$ ($-1 \leq R_i \leq 1$).

The simulated displacement and control current responses generated with the noise are used for the identification. The response signals with 1% and 10% random noise in the time domain are shown in Figure 4. The percentage deviations of the estimated dynamic parameters of the oil-film bearings with different noises are shown in Figures 5 and 6, showing the percentage deviations of the estimated dynamic parameters of the BELM and residual unbalanced force with different noises. In Figures 5 and 6, the horizontal axis represents the estimated parameters, and the vertical axis shows the percentage error between the estimated values and the assumed values. It can be seen that the estimated values are close with the assumed values even with the addition of noise, which indicates that the identification errors for the different levels of noise are negligible. Therefore, the identification method has high accuracy and good robustness to measurement noise.

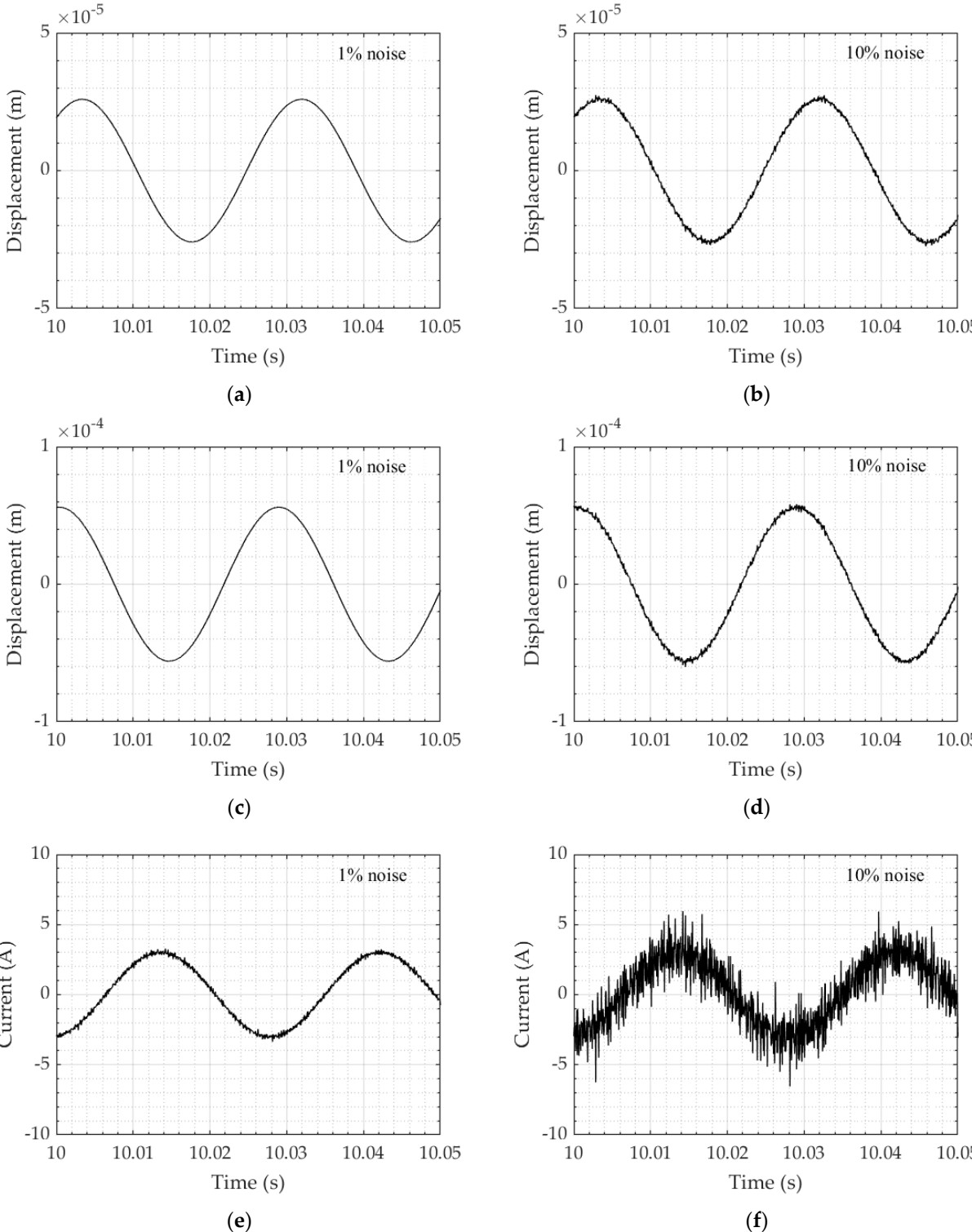

**Figure 4.** The response signal with 1% and 10% random noise. (**a**) Displacement of bearing 1 in *x*-direction with 1% noise; (**b**) displacement of bearing 1 in *x*-direction with 10% noise; (**c**) displacement of BELM in *x*-direction with 1% noise; (**d**) displacement of BELM in *x*-direction with 10% noise; (**e**) control current for BELM in *x*-direction with 1% noise; (**f**) control current for BELM in *x*-direction with 10% noise.

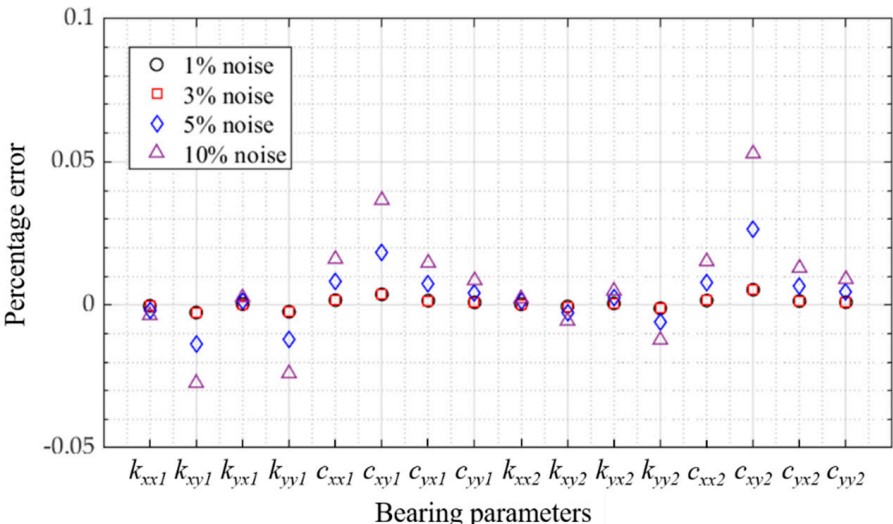

**Figure 5.** Percentage deviations of the estimated bearings dynamic parameters with the addition of noise.

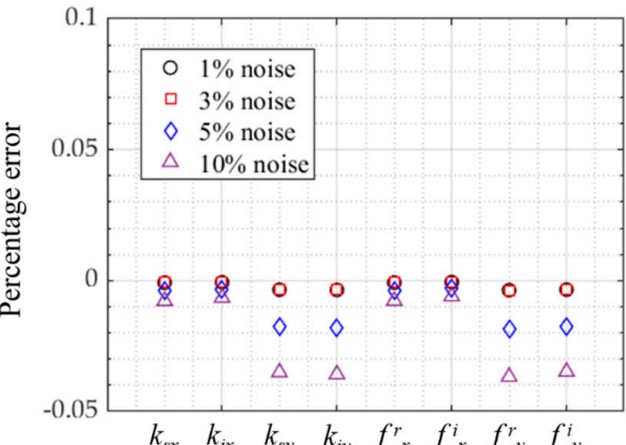

**Figure 6.** Percentage deviations of the estimated BELM dynamic parameters and the residual unbalanced force with the addition of noise.

## 5. Experimental Identification and Verification

### 5.1. Experiment Description

The experimental test rig [4] is shown in Figure 7. The numerical model is the same as the model described in the numerical simulation section. Two cylindrical oil-film bearings are placed at each end of the rotor. A consequent-pole bearingless motor is placed in the middle between the two bearings. The stator of this bearingless motor has motor windings and suspension windings. The suspension windings are used for levitation, which are three phases and have two poles. The motor windings are used for rotation, which are three phases and have eight poles. The control of the BELM is carried out using a DSP system with a sampling frequency of 20 kHz, and the driving current is generated by PWM amplifiers. Two pairs of eddy-current displacement sensors are placed on both sides of the bearings and the BELM. The displacement response of the bearings and the BELM are obtained by averaging the displacement signals from each set of two displacement sensors. A non-contact encoder at the end of the shaft is used to detect the rotational angle of the rotor, which gives a reference for the phase information for the shaft displacement and control current signals.

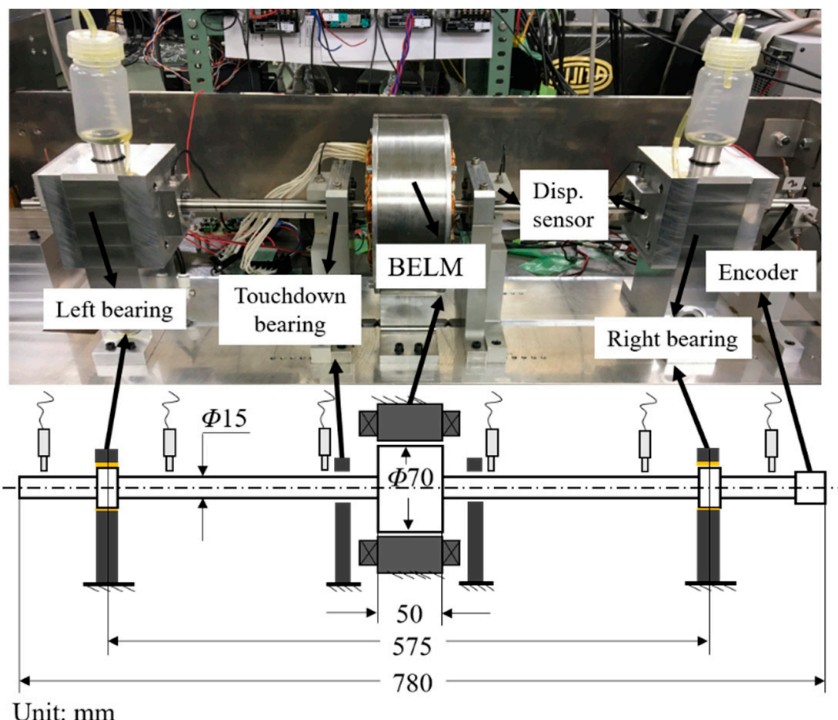

**Figure 7.** The experimental test rig.

Before the dynamic reduction of the rotor model, a modal test experiment was carried out to verify the accuracy of the model, as shown in Figure 8. Table 5 shows the measured and simulated free–free natural frequencies of the rotor. The simulated modes matched the measured ones very well, within a maximum relative error of 2%.

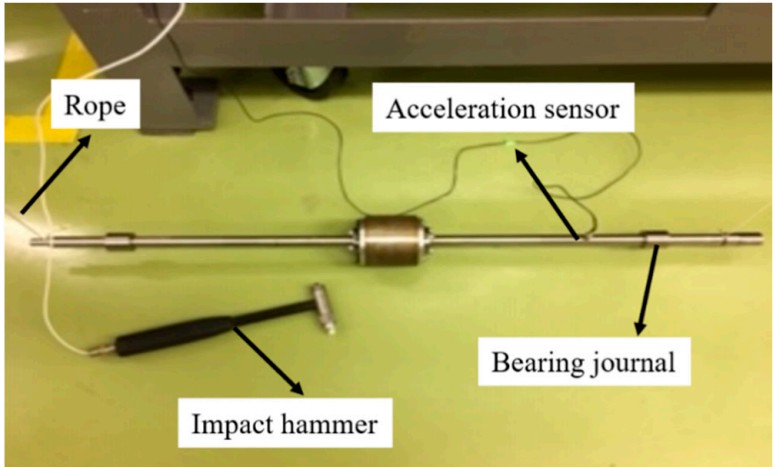

**Figure 8.** Modal test.

**Table 5.** Simulated and measured modal frequencies.

| Modal | Measured Frequency | Calculated Frequency | Error |
|:---:|:---:|:---:|:---:|
| 1 | 83.3 Hz | 82.9 Hz | −0.48% |
| 3 | 510.8 Hz | 507.5 Hz | −0.64% |
| 4 | 770.5 Hz | 756.5 Hz | −1.82% |

In the experiment, the reference position of the motor center is defined by a zero-power controller. When the control current through the suspension windings is zero, the motor center is defined as $X_{ref} = 0$ (mm), $Y_{ref} = 0$ (mm).

Since the oil-film dynamic parameters change with the rotational speed, the rotational speed of the rotor system is fixed at 2100 rpm to simplify the problem. Selection of the $P$ coefficients is required to have a noticeable impact on the response of the rotor. Due to the sensor resolution and noise of the measurement system, the change in rotor response may not be observed when the difference between the two $P$ coefficients is too small. To obtain appropriate data for identification, different $P$ coefficients were applied to obtain different rotor responses. The $P$ coefficients used in our experiment were 52,000, 54,000, and 56,000. The integral, derivative, and filter coefficients were the same as the coefficients in Table 2. The measured displacements of the bearings and the BELM and the control current of the BELM with different $p$ values at 2100 rpm are shown in Figure 9. The amplitudes of the bearing and the BELM decrease as the $P$ coefficient is increased because increasing $P$ means the stiffness of the rotor system is increasing. Similarly, as the amplitude of the BELM decreases, the control current also decreases.

### 5.2. Experiment Results

To validate the proposed method, three groups of PID control parameters were applied to the BELM, and each group had two sets of $P$ coefficients in the $x$ and $y$ directions to obtain two linearly independent responses of the rotor system, as shown in Table 6. Each group of the response data was collected three times to ensure the repeatability of the identification.

**Table 6.** Groups of PID control parameters.

| Directions | Group1 ($G_1$) | | Group2 ($G_2$) | | Group3 ($G_3$) | |
|:---:|:---:|:---:|:---:|:---:|:---:|:---:|
| | **Set 1** | **Set 2** | **Set 1** | **Set 2** | **Set 1** | **Set 2** |
| $x$ | 52,000 | 54,000 | 52,000 | 56,000 | 54,000 | 56,000 |
| $y$ | 54,000 | 52,000 | 56,000 | 52,000 | 56,000 | 54,000 |

Figure 10 shows the identified stiffness and damping parameters of the oil-film bearings. $G_{m,n}$ is the group of PID control parameters, where $m$ is the group number, and $n$ is the number of the test. The direct stiffness parameters $k_{xx}$ and $k_{yy}$ of bearing 1 (B1) and bearing 2 (B2) are close. The direct damping parameters $c_{xx}$ and $c_{yy}$ of B1 are a little smaller than B2. For each bearing, the direct stiffness parameters $k_{xx}$ and $k_{yy}$, and the damping parameters $c_{xx}$ and $c_{yy}$ are very close. Therefore, the bearings of the test rig are almost isotropic. As for the cross-coupled parameters, it is apparent that the cross-coupled stiffness and damping parameters have the opposite sign. The cross-coupled stiffness $k_{xy}$ and $k_{yx}$ are opposite in the two bearings. Since the test rig is nearly symmetrical and the experimental conditions for the two bearings are the same, it may be caused by misalignment of the rotor system or by deformation of the flexible rotor. To verify that the identified parameters are reasonable, comparisons are made between the rotor unbalanced responses estimated by the identification values and the measured results in the next section.

Figure 11 shows the identified displacement-force parameters, current-force parameters, and the residual unbalanced forces. The experimental results are found to cover a small range, although the PID control parameters are different. The slight variation could be due to the lubrication state of the oil-film bearings, the experimental test conditions, or modeling errors. The reliability of the identified parameters is discussed and verified in the next section.

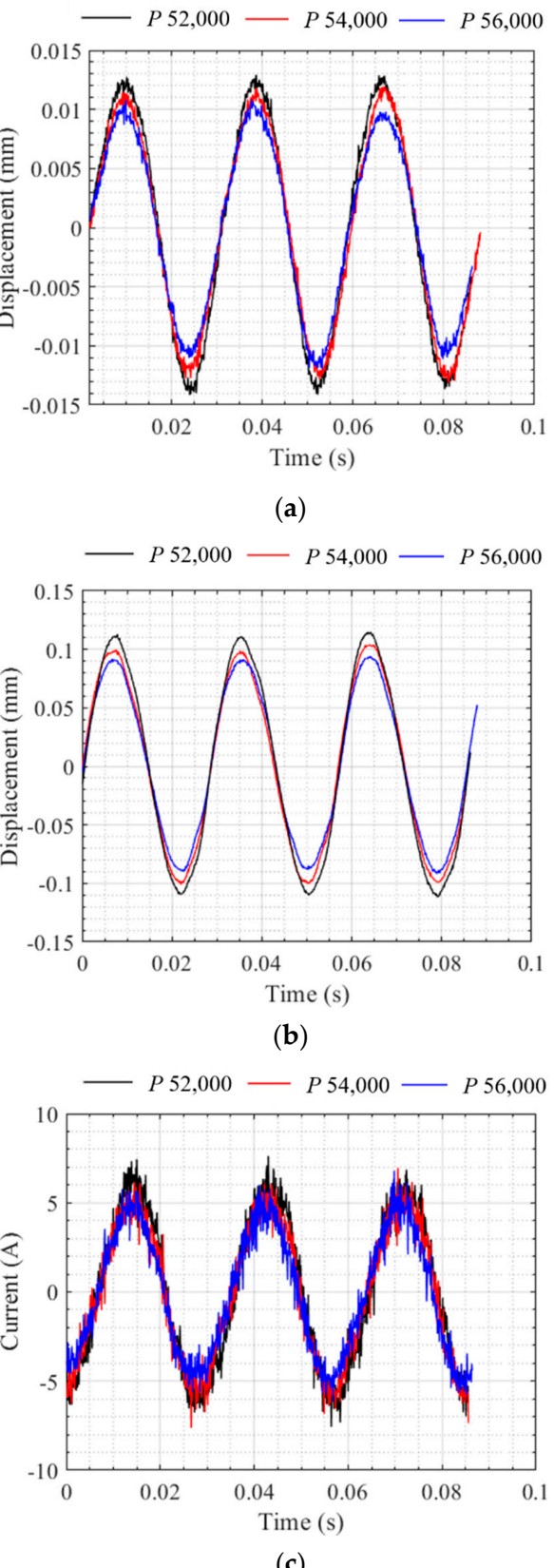

**Figure 9.** Measured data with different *p* values. (**a**) Displacement of bearing 1 with different *p* values in the *x*-direction; (**b**) displacement of BELM with different *p* values in the *x*-direction; (**c**) control current for the BELM with different *p* values in the *x*-direction.

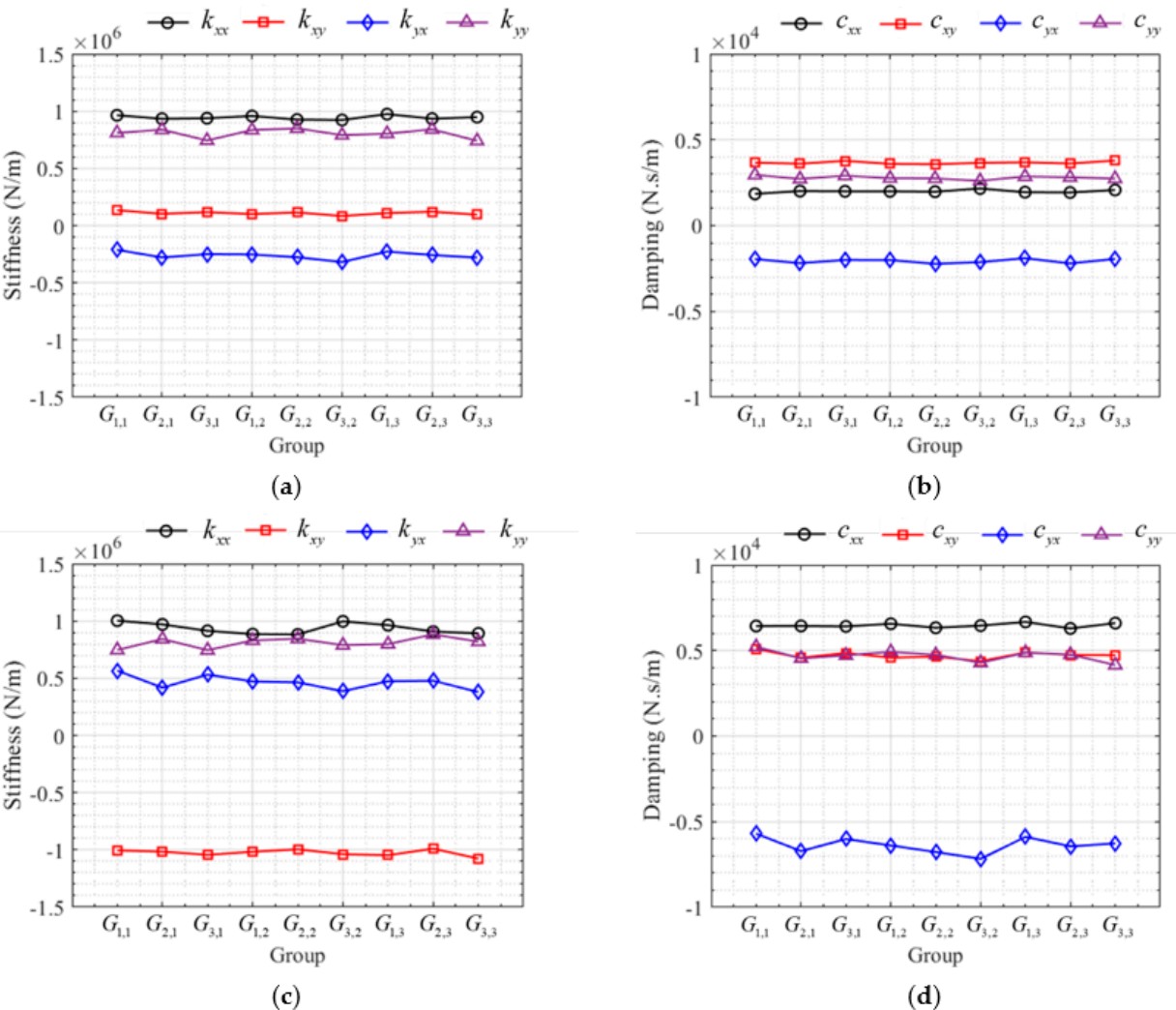

**Figure 10.** Identified dynamic parameters of the bearings. (**a**) Identified stiffness parameters of bearing 1; (**b**) identified damping parameters of bearing 1; (**c**) identified stiffness parameters of bearing 2; (**d**) identified damping of bearing 2.

*5.3. Verification of the Results*

To ensure that the identified parameters are reasonable, verification is indispensable. A vibration suppression test was carried out to verify the unbalanced force identification results. Moreover, comparisons are made between the frequency response characteristics and the rotor unbalanced responses estimated by the identification values and the measured results.

5.3.1. Vibration Suppression Test

When the rotor rotates at a speed of 2100 rpm, the residual unbalanced force leads to whirling of the rotor. According to Section 5.1, the magnitude and phase of the residual unbalanced forces in the $x$ and $y$ directions were obtained. If the opposing force can suppress the vibration of the rotor, it can be proved that the identified residual unbalanced force is correct. Figure 12 shows the vibration control system. A force in the opposite direction to the residual unbalanced force can be generated by injecting currents $I_{sx}$ and $I_{sy}$ into the suspension coil.

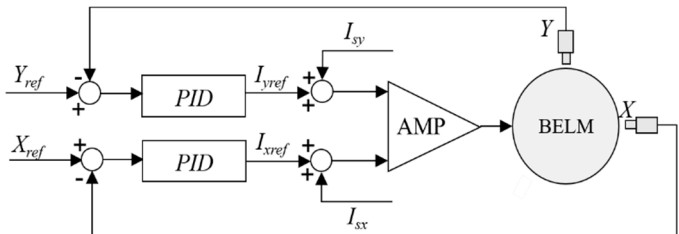

**Figure 11.** Identified parameters of the BELM and residual unbalanced force. (**a**) Identified current-force parameters of BELM; (**b**) identified displacement-force parameters of BELM; (**c**) identified residual unbalanced force.

**Figure 12.** Block diagram for the vibration suppression control system.

The current injected into the suspension coil can be expressed as

$$I_{sx} = \frac{f_x}{k_{ix}}\cos(\omega t + \varphi_x + \pi)$$
$$I_{sy} = \frac{f_y}{k_{iy}}\cos(\omega t + \varphi_y + \pi)$$

$$(17)$$

where $I_{sx}$ and $I_{sy}$ are the suppression current in the *x* and *y* directions; $f_x$ and $f_y$ are the magnitudes of the suppression current, and $\varphi_x$ and $\varphi_y$ are the phases of the suppression current, respectively.

In the vibration suppression test, the *P* coefficients of the PID controller used in the *x* and *y* directions were 52,000 and 54,000, respectively, and the vibration suppression result is shown in Figure 13. It is clear that before injecting the suppression currents, the vibration amplitudes in the *x* and *y* directions are different. Since the *P* coefficient in the *x*-direction is smaller than the *P* coefficient in the *y*-direction, the amplitude in the *x*-direction is greater than the amplitude in the *y*-direction. When the suppression current is injected in the third second, the amplitude of vibration is suppressed, and the amplitudes in both directions are reduced to approximately 0.01 mm. The results show that adding the force in the opposite direction to the residual unbalanced force can suppress the vibration of the rotor, proving that the identified residual unbalanced force is correct.

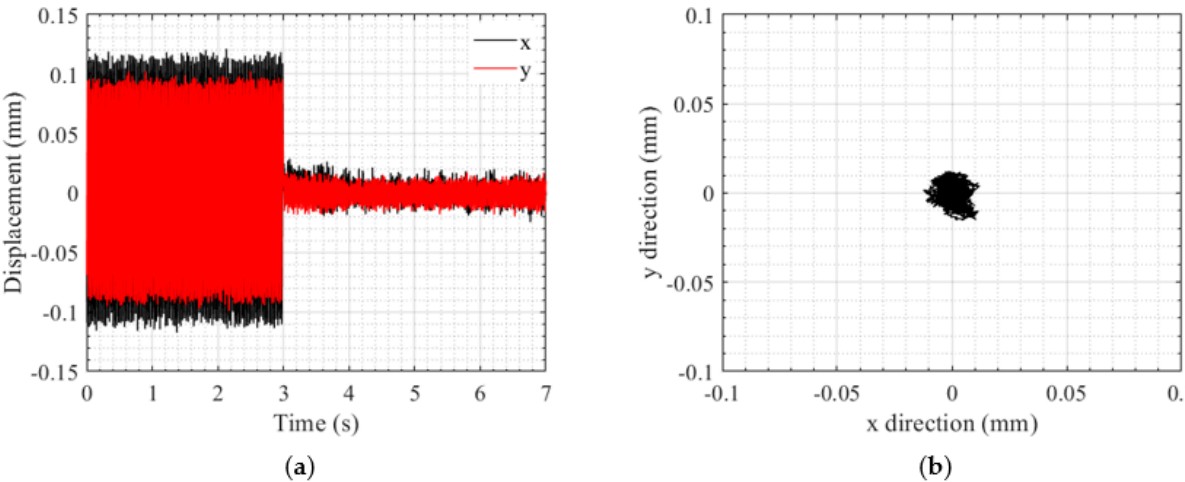

**Figure 13.** Vibration suppression test results. (**a**) BELM response before and after suppressing the vibration; (**b**) vibration suppression results.

### 5.3.2. Frequency Response Characteristics

Figure 14 shows the measured and estimated frequency response characteristics from the excitation current to the displacement of the BELM at 2100 rpm. The measurement system is the same as that used for the suppression control system in Figure 12. The suppression current is replaced by the excitation current. The magnitude of the excitation current is 0.8 A. The excitation frequency range is set from 1 Hz to 200 Hz. The numerically estimated results are calculated from the rotor model. The identified unbalanced forces, bearing dynamic parameters, and BELM dynamic parameters are added onto the corresponding nodes.

Figures 14–16 illustrate the frequency responses in the *x* and *y* directions when the *P* coefficients of the PID controller in the *x* and *y* directions are 52,000, 54,000, and 56,000, respectively.

The black and red lines are the measured and estimated frequency responses, respectively. It is seen that the estimated results are in good agreement with the measured results despite the PID control parameters being different.

Furthermore, there are other peaks at multiple frequencies in addition to the peak at the same frequency as the rotational speed. The vibration at these frequencies may be due to an electrical disturbance force in the motor. Waterfall plots under the excitation are shown in Figure 17. The results show that vibration at these multiple frequencies is always present in the excitation process, which indicates that the excitation current does not induce this vibration.

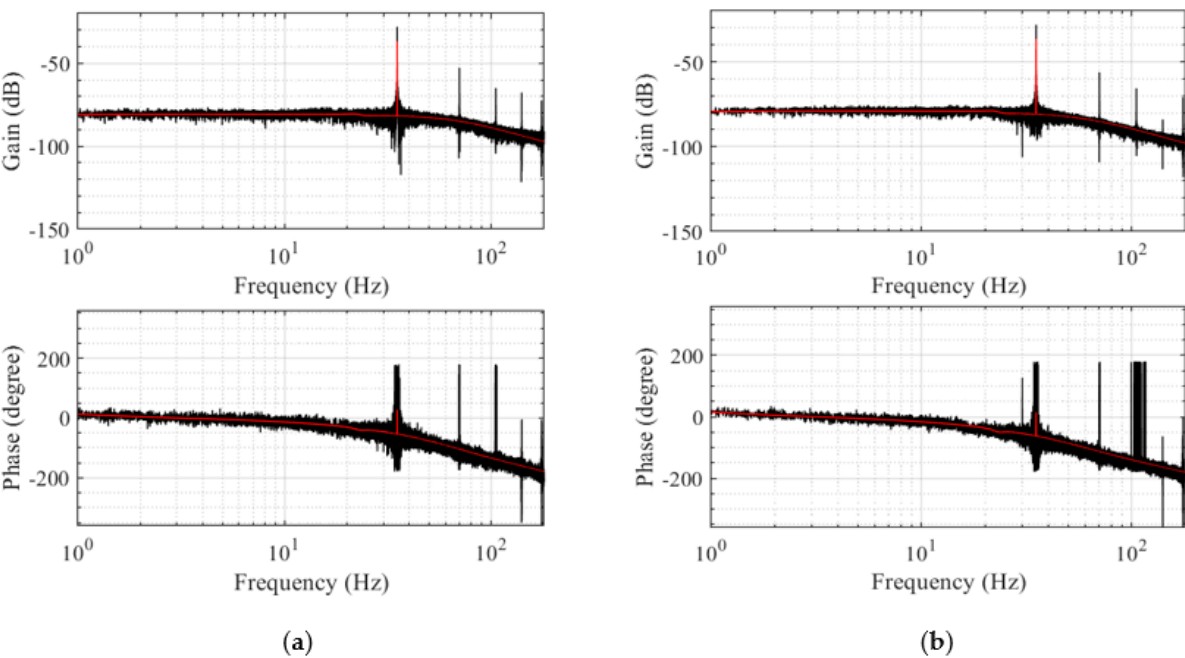

**Figure 14.** Measured and estimated frequency responses in the *x* and *y* directions (*P* = 52,000). (**a**) Frequency response in the *x*-direction; (**b**) Frequency response in the *y*-direction.

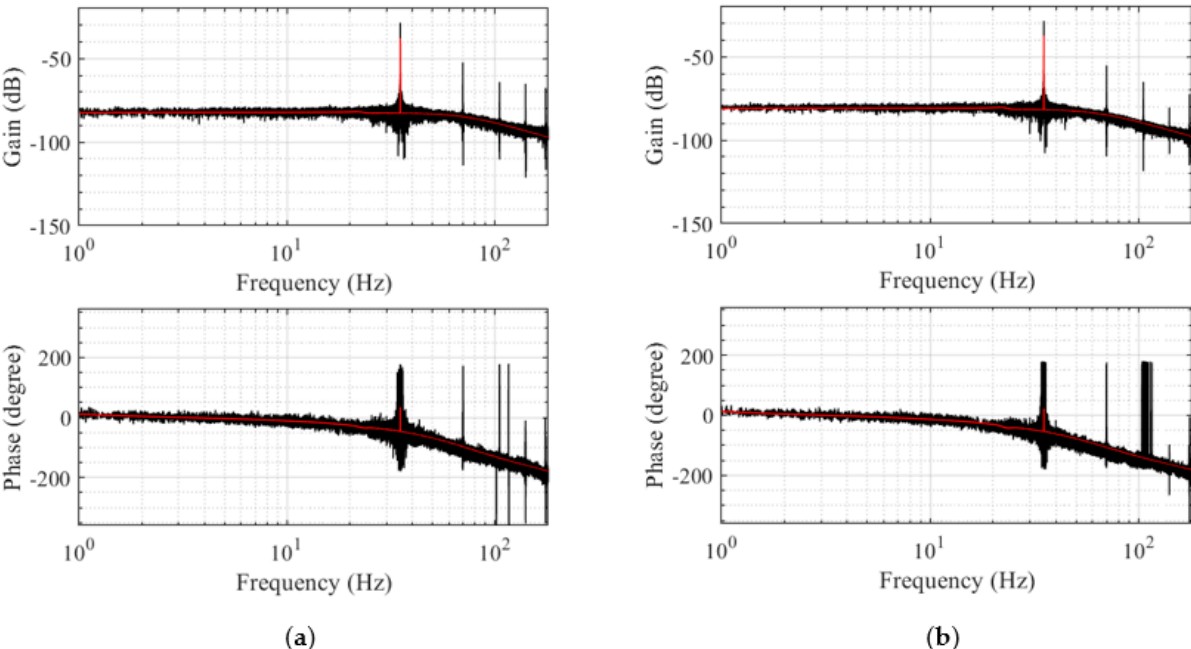

**Figure 15.** Measured and estimated frequency responses in the *x* and *y* directions (*P* = 54,000). (**a**) Frequency response in the *x*-direction; (**b**) frequency response in the *y*-direction.

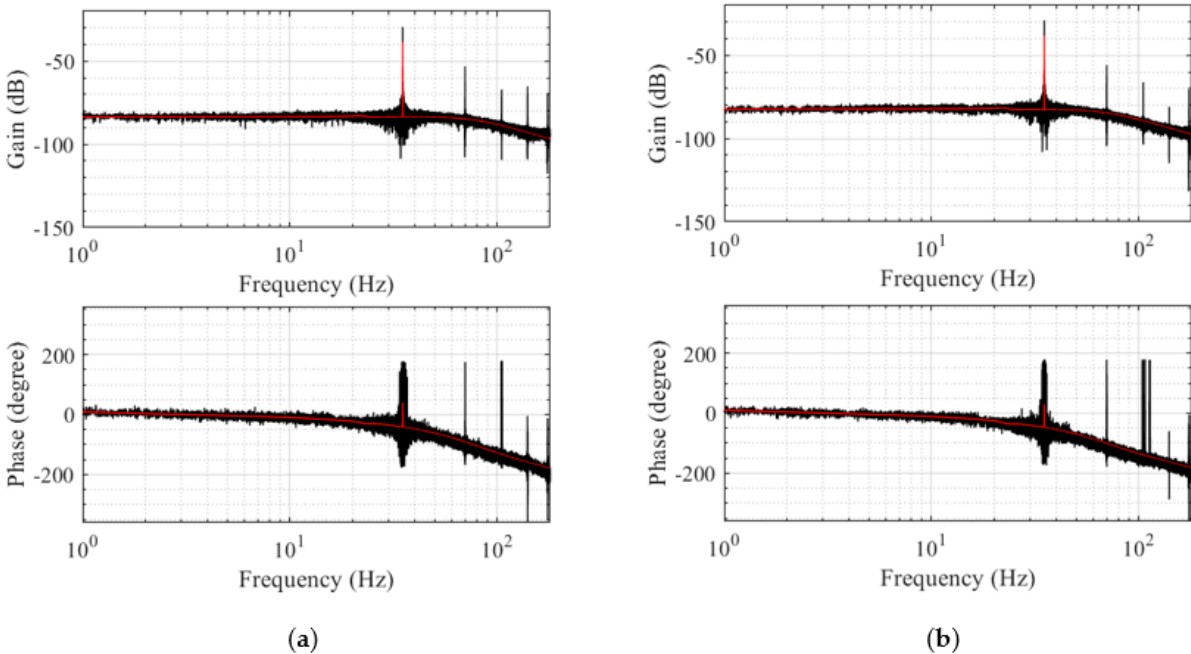

**Figure 16.** Measured and estimated frequency responses in the *x* and *y* directions (*P* = 56,000). (**a**) Frequency response in the *x*-direction; (**b**) frequency response in the *y*-direction.

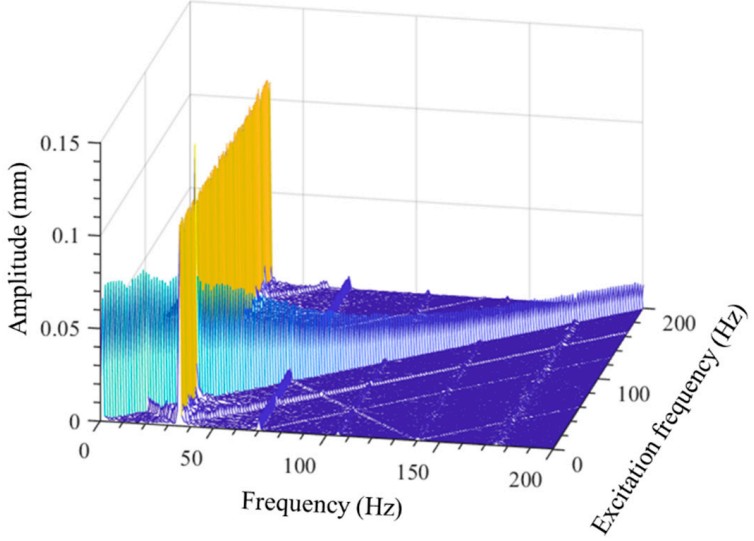

**Figure 17.** Waterfall plot at 2100 rpm.

### 5.3.3. Unbalanced Response

The FE model before reduction is used to estimate the unbalanced response of the rotor. Comparisons are made between the rotor unbalanced responses estimated by the identified values and the measured results at the displacement sensors locations.

Figure 18 shows the results of comparison when the *P* coefficients of the PID control parameters in the *x* and *y* directions are 52,000 and 54,000. Figure 19 shows the results of comparison when the *P* coefficients of the PID control parameters in the *x* and *y* directions are 54,000 and 56,000. In Figures 18 and 19, the solid black and red lines are the real parts of the estimated responses of all the nodes of the rotor in the *x* and *y* directions. The black and red scattered round points are the real parts of the measured responses at the displacement

sensor locations. The dashed lines and scattered diamond points are the imaginary parts of the estimated and measured responses.

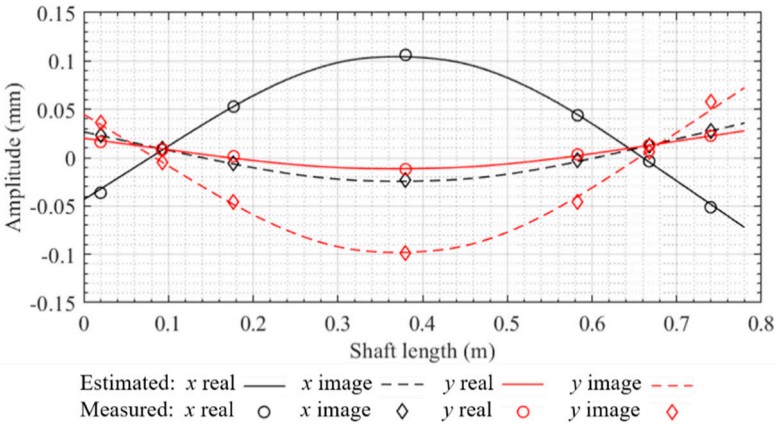

**Figure 18.** Comparisons between measured and estimated vibration response when the *P* coefficients in the *x* and *y* directions are 52,000 and 54,000.

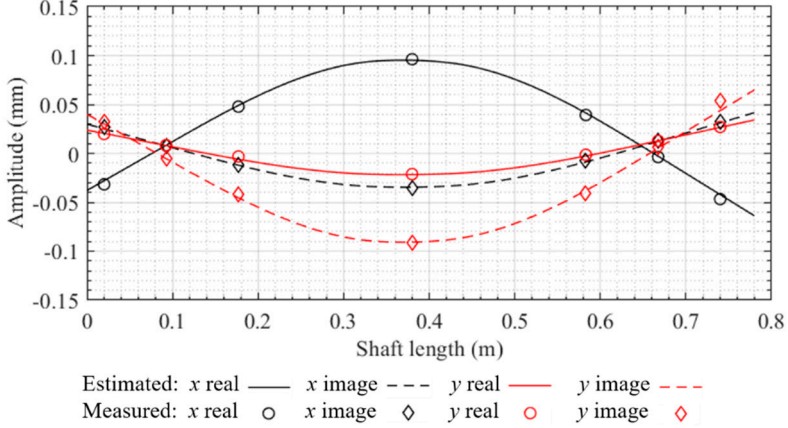

**Figure 19.** Comparisons between measured and estimated vibration response when the *P* coefficients in the *x* and *y* directions are 54,000 and 56,000.

According to the results, the measured real and imaginary parts of the responses match well with the estimated results. The good consistency between the experimental and estimated responses shows that the dynamic parameters of the rotor system were reasonably identified.

## 6. Conclusions

This study presents a method for dynamic parameters of the BELM, oil-film bearings, and the residual unbalanced forces identification in a flexible rotor system. The measured rotor unbalanced responses and control currents under different PID controller coefficients and the FE rotor model are used for identification. The effectiveness of the method is verified by numerical simulation and experiments. The numerical simulation results show that the proposed method can against measurement noise and that the identification results are accurate. The proposed method was also applied to a flexible rotor test rig driven by a BELM. Three groups of PID control parameters were investigated, and for each group, the response data were collected three times. The results of the different tests are close together. To confirm the experimental identification results are correct, verification experiments were carried out. By applying a force in the opposite direction to the identified residual unbalanced force to the rotor, the vibration amplitude of the rotor was successfully

suppressed. The frequency response characteristics and unbalanced responses of the rotor estimated by the values of the parameters identified show good consistency with the measured results. The verification results indicate that the proposed method is effective and that the identification results are reliable.

**Author Contributions:** Conceptualization and methodology, Y.C.; software, validation, formal analysis and investigation, Y.C. and R.Y.; writing—original draft preparation, Y.C. and N.S.; writing—review and editing and supervision, T.S. and J.M.; All authors have read and agreed to the published version of the manuscript.

**Funding:** NSK foundation for the advancement of mechatronics.

**Institutional Review Board Statement:** Not applicable.

**Informed Consent Statement:** Not applicable.

**Data Availability Statement:** Not applicable.

**Acknowledgments:** The authors would like to acknowledge the support from the China Scholarship Council (No.201906280434).

**Conflicts of Interest:** The authors declare no conflict interest.

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
