# Peer review of "Identification of Bearing Dynamic Parameters and Unbalanced Forces in a Flexible Rotor System Supported by Oil-Film Bearings and Active Magnetic Devices"

_actuators, doi:10.3390/act10090216_

Round 1

Reviewer 1 Report

  1. The authors have a very nice test rig, with the potential to make some good contributions to the field of rotating machinery.
  2. However, it is not clear from the paper what is unique about their work compared with earlier work in the literature (as detailed in item 3 below). Additionally, the authors begin the paper by addressing oil whip and oil whirl, which are both phenomena associated with rotor-bearing instability that presents itself as sub-synchronous vibrations.  However, the paper only seems to address synchronous vibrations induced by unbalance.  It seems as if the authors are just showing that they can successfully predict the response of their test rotor, which is common in rotor dynamics community.  Therefore, clarification of what they are actually doing, as well as clarification as to what specific unique contributions they are adding to the literature must be done before paper is publishable.
  3. While the authors are doing nice work, they are repeating earlier studies that are not referenced and should be. Their comment in the introduction at lines 106-108 is not correct and needs to be removed or clarified.  In particular, the authors have overlooked much of the work by Kasarda, et. al. that deals with utilizing an active magnetic bearing (sometimes referred to as an active magnetic damper) in conjunction with conventional support bearings to improve synchronous and/or subsynchronous (i.e. stability) vibrations in a rotor-bearing system.  More specifically:
    1. The authors are not the first to propose using an active magnetic bearing in conjunction with conventional bearings to stabilize rotating machinery by applying forces to reduce non-synchronous rotor vibrations. The authors need to read the following paper, include a comment regarding it and how their work is different, and include it as a reference:  Kasarda, M.E., Mendoza, H., Kirk, R. G., and Wicks, A., “Reduction of Subsynchronous Vibrations in a Single Disk Rotor Using An Active Magnetic Damper,”  Mechanics Research Communications, 31, Issue 6, November-December 2004, pp689-695.
    2. The authors are not the first to propose using an active magnetic bearing in conjunction with conventional bearings to modify/control rotor dynamic properties of a synchronous rotor response in conventional bearings. The authors need to read the following paper, include a comment regarding it and how their work is different, and include it as a reference:  Kasarda, M. E. F., Allaire, P. E., Humphris, R. R., Barrett, L. E., "A Magnetic Damper for First-Mode Vibration Reduction in Multimass Flexible Rotors,"  Journal of Engineering forGas Turbines and Power, ASME, Vol. 112, No. 4, October 1990, pp. 463-469.
    3. Similarly, the authors are not the first to propose a system identification technique for measurement of shaft forces using Active Magnetic Bearings. The authors need to read the following paper, include a comment regarding it and how their work is different, and include it in the references:  Marshall, J., Kasarda, M., and Imlach, J., “A Multi-Point Measurement Technique For The Enhancement of Force Measurement with Active Magnetic Bearings,” ASME Journal of Engineering for Gas Turbine and Power, Vol. 125, No. 1, January 2003, pp. 90-94
    4. “Active balancing,” were rotating forces are applied to magnetic bearings to counteract unbalance forces has been done for years.  What makes your work unique?
    5. Additionally, the authors leave out other notable references, including work by Nordman at Darmstadt, as well as Swanson and Kirk at Virginia Tech, for their respective work in measuring dynamic coefficients of fluid-film bearings using test rigs where an active magnetic bearing is used to apply loads.
    6. Please also clarify what is different in this work from your previous publications (references 3-6)
  1. As shown in Figure 2, the authors appear to be using assumed values to determine an equation of motion and a forced response solution to use for their regression analysis. They then appear to be using the regression analysis to determine new values, and then using these for comparison with their original assumed values.  This doesn’t make sense as described because the analysis is circular, and it is implied that the extracted values should completely match up to the assume values if no additional input is provided.  This must be explained more clearly as currently it looks like an invalid approach, where you are predicting your assumed values.  ???   Please explain.
  2. Why are you choosing different controller P values for X and y directions? What is meant by “set 1” and “set 2” in Table 6?
  3. You state that some of your cross-coupled k and c values are of opposite signs. This is not an uncommon phenomena in rotor-dynamics, and depending on those signs can be the literal cause of instability issues.  The paper would be strengthened by discussing the cross-coupling parameters and sign differences briefly, rather than not having an explanation for the sign difference at all.
  4. What speed is the rotor spinning at for the results In Figure 14? Where are these results measured?
  5. Which is the measured response and which is the predicted response in Figures 18 and 19?
  6. The location of displacement sensors in Figure 7 do not line up with points presented in Figures 18 and 19.
  7. Typos must be corrected.

Overall, I want to encourage the authors to clarify their work (including what the unique contributions are) and resubmit.  It is a very nice test rig, and there may be some very good contributions here, but as written this just isn't clear.  

Author Response

Thank you very much for your kind review. Please see the attachment including the revised article and the answer sheet.

Reviewer 2 Report

The rotor supported by oil-film bearing is easy to produce the oil whip and oil whirl. The author proposed the use of active magnetic bearing / bearingless motors to decrease the vibration and stabilize these systems. The stiffness and damp of the oil-film bearing, the current-force and displacement-force coefficent of the AMBs/BELMs, the residual unbalaced force are the foundation for the rotor dynamics analysis and optimization of the control strategy. So the author propose a method to simultaneously identify these parameters of the oil-film bearing and AMBs/BELMs along with the residual unbalanced forces during the unbalanced vibration of the rotor. The numerical simulation and experimental study are carried out to verify the proposed method. The paper is well organized and creative, but some places need to be modified and improved.
1. It is not clear which the AMBs/BELMs is used. The oil-film bearing needs eccentric oil-film to work, the central position the AMBs with feed control is fixed. When these bearings work at the same time, how to coordinate the axial position of the oil-film bearings and AMBs?
2. The rotational speed of the rotor system is fixed at 2100rpm in the experimental study, but the rotational speed is 1800rpm in the numerical simulation. Please explain why the rotational speed is different in the two cases.

Author Response

(The authors gave the same response as above.)
